# Familiarity Processing through Faces and Names: Insights from Multivoxel Pattern Analysis

**DOI:** 10.3390/brainsci14010039

**Published:** 2023-12-30

**Authors:** Ana Maria Castro-Laguardia, Marlis Ontivero-Ortega, Cristina Morato, Ignacio Lucas, Jaime Vila, María Antonieta Bobes León, Pedro Guerra Muñoz

**Affiliations:** 1Department of Cognitive and Social Neuroscience, Cuban Center for Neurosciences (CNEURO), Rotonda La Muñeca, 15202 Avenida 25, La Habana 11600, Cuba; castro.laguardia@gmail.com (A.M.C.-L.);; 2Mind, Brain and Behavior Research Center (CIMCYC), University of Granada (UGR), Avda. del Hospicio, s/n P.C., 18010 Granada, Spainjvila@ugr.es (J.V.);

**Keywords:** faces, familiarity processing, multivoxel pattern analysis

## Abstract

The way our brain processes personal familiarity is still debatable. We used searchlight multivoxel pattern analysis (MVPA) to identify areas where local fMRI patterns could contribute to familiarity detection for both faces and name categories. Significantly, we identified cortical areas in frontal, temporal, cingulate, and insular areas, where it is possible to accurately cross-classify familiar stimuli from one category using a classifier trained with the stimulus from the other (i.e., abstract familiarity) based on local fMRI patterns. We also discovered several areas in the fusiform gyrus, frontal, and temporal regions—primarily lateralized to the right hemisphere—supporting the classification of familiar faces but failing to do so for names. Also, responses to familiar names (compared to unfamiliar names) consistently showed less activation strength than responses to familiar faces (compared to unfamiliar faces). The results evinced a set of abstract familiarity areas (independent of the stimulus type) and regions specifically related only to face familiarity, contributing to recognizing familiar individuals.

## 1. Introduction

In daily life, the brain combines sensory inputs such as faces, written proper names, and semantic knowledge to recognize familiar individuals. Several studies and theoretical models have recognized the existence of specific neural systems for processing the structural properties of faces and written words (including written proper names). In the case of faces, a “core system” has been described, comprising regions such as the occipital face area (OFA), the fusiform face area (FFA), and the superior temporal sulcus (STS-FA) that are specialized in the processing of facial invariant features [1,2,3]. Analogously, it has also been reported that the visual word fusiform area (VWFA) plays a fundamental role in the processing of the structural aspects of written words, which is relevant to the low-level properties of written proper names [4,5].

The above-mentioned areas are connected to a set of other regions processing semantic information [1,2,3]. These areas are considered to be more abstract, which means that they process familiarity and other dynamical aspects of the stimulus independent of its modality. Faces are the more studied source of information on familiarity recognition. Familiarity is defined as a form of explicit or declarative memory entailing the retrieval of multimodal identity-specific knowledge about a person from long-term memory [1,6]. This type of memory involves the ability to recollect relevant biographical and semantic facts (profession, name, hobbies, specific personal encounters) together with the emotional responses to the person, which depend on many factors, including the length of time spent with the person, number of previous encounters, duration and quality of the relationship, or information accumulated about the person.

Neuroimaging studies on face/name identity recognition have used a variety of tasks to examine familiarity [7]. Some studies have focused on the cognitive processing of familiarity using tasks that require explicit or implicit recognition (i.e., judging whether a face/name was familiar or unfamiliar, or face/name gender classification). However, a number of neuroimaging studies on face/name identity recognition have also used a category of faces/names called personally familiar which included the faces/names of family members and friends such as the romantic partner, parents, or children of the participants [8,9,10,11,12]. This type of identity recognition task involves stronger familiarity not only in terms of biographical/semantic knowledge but also in terms of emotion.

Models describing how our brain recognizes faces propose the so-called “extended system”. This system comprises neural areas associated with the retrieval of personal information and the emotional response associated with the visualization of a familiar face [1,3,13]. However, there is evidence pointing out the existence of regions specialized in face familiarity processing and also the salience of faces for recognizing familiarity that are different from those related to proper names. For example, in an early study by Ellis, Quayle, and Young (1999), faces produced significantly larger Skin Conductance Responses (SCRs) than names. No differences were found in SCRs between familiar and unfamiliar names [14].

Some studies using lesion-based data support the idea of a modality-specific semantic system. Those studies have found modality-specific deficits for famous faces and names as a function of the laterality of the lesion [3,10,11,12]. Eslinger et al. (1996) also found, in two postencephalitic subjects (EK and DR) with a different pattern of lesion in the left and right temporo-occipital areas, respectively, a different pattern of impairment in the processing of faces and names. EK, the patient with a lesion in the left temporo-occipital areas, exhibited a considerable loss of proper names knowledge, while patient DR, with a right-sided lesion, showed a marked difficulty in recognizing faces [15]. Gainotti (2012) extensively reviewed both case and group studies examining the effect of lateralized left or right temporal lobe lesions on the recognition of famous people. The author concluded that the data were most consistent with a modality-specific proposal [16].

Also, early neuroimaging studies using PET [17] and fMRI [18] identified distinct but sometimes overlapped activations for faces and names in familiarity processing, suggesting the access to a common familiarity knowledge system after a pre-semantic processing stage. Activation patterns, regardless of stimulus modality, involved extensive left hemisphere networks and bilateral activity in the hippocampus, posterior cingulate, and middle temporal gyrus [17,18]. Famous faces activated the right hemisphere and famous names were more left- lateralized. These studies indicate that a set of shared regions typically associated with the retrieval of biographical knowledge and social–affective processing appears to participate in familiarity processing regardless of the modality of presentation. However, studies using activation-based methods may not be the optimal way to address this problem. Importantly, univariate statistical models do not encode relationships between voxels [19,20]. Instead, each voxel activation value is modeled separately, to detect brain regions that respond more strongly during one experimental condition than during another. Also, in classical brain mapping analyses, the data are typically spatially smoothed to focus sensitivity on the overall activations of functional regions. Therefore, population-code information reflected in subtle differences between nearby voxels, such as the difference between an unfamiliar and a familiar face (or between an unfamiliar and a familiar name), may be lost [19].

Alternatively, in multivariate analysis, multiple responses are jointly tested for differences between experimental conditions. This approach has opened new questions and ways of thinking about the neural representation of familiarity, such as those from the study from Duchaine and Yovel (2015), which were difficult or even impossible to formulate from a traditional univariate point of view. While classical activation analysis is aimed at revealing a region’s involvement in a certain cognitive function, MVPA, on the contrary, pursues looking into each region and revealing its “representational content” by testing for combinatorial effects [13].

Our experimental design focused on assessing neural responses to familiar persons with a high degree of familiarity; specifically, individuals with whom the participants shared close relationships. While this approach provides valuable insights into the neural mechanisms underlying familiarity processing, we acknowledge the importance of extending our investigations to encompass varying degrees of familiarity. Future studies could explore neural responses to persons of varying familiarity levels, including less familiar or famous persons, to elucidate the nuanced nature of familiarity processing. The consideration of a broader familiarity spectrum will contribute to a more comprehensive understanding of the neural underpinnings of person recognition.

In our study, we address the question of how our brain processes familiarity from faces and names. We employed searchlight multivoxel pattern analysis (MVPA) to investigate two contrasting hypotheses. Firstly, we explored whether multivoxel patterns associated with familiarity could be detected in distinct brain regions, as indicated by the precise decoding of familiarity for both faces and names. Some of these patterns may reveal shared characteristics between the two modalities, supported by successful cross-decoding from faces to names. Secondly, we examined the possibility that familiarity could be accurately decoded for a particular modality in specific regions where common patterns do not exist, leading to cross-decoding failures.

## 2. Materials and Methods

### 2.1. Participants

The sample consisted of 32 healthy volunteers (12 males), aged between 18 and 35 (mean = 21.4, SD = 4.09). All participants, and the faces of the people in the pictures provided by them, were Caucasian. All of them have an educational level above high school. None reported current physical or psychological problems, and none were under pharmacological treatment.

All participants had a highly positive relationship with their father, mother, romantic partner, and best friend (the sex of the best friend is the opposite of the romantic partner). A Likert scale of 1 to 5 was used to measure the relationship quality (positive affect) with their loved ones. They had to report a score of 4 or 5 for all the loved ones. To control the familiarity, the participants had to have lived with their parents until the age of 18, have a romantic relationship of minimum 6 months and maximum 6 years and not live together. As control images, photographs of the loved ones of another participant were presented. In addition, participants had to provide recent pictures of their loved ones. All participants provided written informed consent for participating in the study protocol and received course credits. The study was approved by the Ethics Committees of the University of Granada and of the Cuban Center for Neurosciences, respectively. The experimental session was carried out at the University of Granada.

### 2.2. Stimuli and Experimental Procedure

Four different categories of stimuli were used: familiar faces, unfamiliar faces, familiar names, and unfamiliar names. These categories were grouped into two factors: type of stimuli (faces or names) and familiarity (familiar or unfamiliar). Familiar stimuli consisted of face pictures and full names (first name + last name) of significant people for each participant.

The study was conducted in an MRI scanner and comprised a total of 180 trials. Each trial consisted of a fixation cross for 3–5 (mean = 4) seconds followed by one of the stimuli displayed for 2 s. All trials were pseudo-randomly presented to each participant, with the restriction that no more than 2 stimuli of the same category were presented consecutively. The total duration of the experiment was 18 min.

All stimuli were visually displayed inside a plain gray circle in the center of the screen. The faces showed part of the neck and hair and were turned black/white with a similar size and level of brightness. All faces looked directly at the camera and showed neutral emotional expressions. For a more detailed description of the stimuli and experimental procedure see [12].

### 2.3. Image Acquisition

A Siemens 3T Tim Trio MR scanner system with a standard birdcage head coil for signal transmission/reception (MAGNETOM, Siemens, Healthcare, Germany) was used to acquire all images. BOLD-contrast-weighted echo-planar images (EPI) were registered as functional scans consisting of 40 interleaved, axial slices of 2.2 mm thickness (with a gap, of 30%) that partially covered the brain from about −57.2 below to about 57.2 mm above the P-A plane. The in-plane resolution was 3 × 3 mm, with the following parameters: FOV = 210 mm, matrix = 70 × 70; echo time (TE) = 23 ms; TR = 3 s with no time gap; flip angle = 90°. The first five volumes were discarded to allow for T1 equilibration effects. Additionally, an MPRAGE T1-weighted structural image (1 × 1 × 1 mm resolution) was acquired with the following parameters: echo time (TE) = 2.52 ms, repetition time (TR) = 2250 ms, flip angle = 90°, and field of view (FOV) = 256 mm. This yielded 176 contiguous 1 mm thick slices in a sagittal orientation.

### 2.4. Preprocessing Pipeline

Functional imaging data were preprocessed using SPM12. First, outlier functional scans and slices were repaired with the Artifact Repair Toolbox (https://www.nitrc.org/projects/art_repair/ (accessed on 5th May 2018). Then, the images were slice-time corrected, taking the middle slice as reference (using SPM12 phase shift interpolation with the unwarp option) and then realigned to the first image in the session. Each anatomical T1 image was co-registered with the mean EPI, bias-corrected and spatially normalized to MNI space, and segmented into gray matter (GM), white matter (WM), and cerebrospinal fluid (CSF).

To estimate individual trial responses, the LS-S (Least Squares-Separate) approach was used. This allowed for correction of the overlap in time of evoked BOLD signals for consecutive trials in rapid ER designs. This LS-S procedure applied a separate GLM for each trial, which is modeled as the regressor of interest while all other trials were joined into a single nuisance regressor, in addition to the movement parameters.

### 2.5. Univariate Analysis

Statistical parametric maps were generated to explore areas where the activation was significantly different for the conditions of interest. An individual voxel probability threshold of 0.05 using family-wise error correction (FWER) was applied to minimize false positive activation foci from the brain maps. We performed the following t-contrasts of interest: (1) all faces versus all names, and (2) all names versus all faces, to explore the signal intensity for faces and names, respectively.

### 2.6. Searchlight MVPA for Familiarity Information Decoding

Searchlight MVPA [21] provides an interesting approach to finding localized informative regions in the brain, avoiding the need for functional knowledge in the traditional definition of a region of interest (ROI). Typically, in this method, a classifier is trained and tested on a small cerebral region that spans the entire brain. Here, we used searchlight MVPA to decode invariant familiarity information in combination with a cross-classification technique.

In cross-classification [22,23], a classifier is trained on data from one cognitive context and then tested on data from another context. Essentially, if it is possible to accurately predict the stimuli from one category using a classifier trained on stimuli from another category, then inferences can be made about the specific role of a given brain region in representing abstract or invariant information across different cognitive contexts. In our experimental design, cross-classification was used to identify neural representations of familiarity that remained invariant to changes in stimulus modality, and modality invariance to changes in familiarity.

For this analysis, although the experiment consisted of a single session, the functional data were artificially divided into three separate runs (containing 126, 126, and 128 volumes, respectively), and the trial betas (estimated by LS-S) were used as features. The searchlights were defined as spheres with 26 neighbors around each voxel in the grey matter volume. In each iteration of the cross-validation (leave-one run out), a fast Gaussian Naïve Bayes classifier [24] was used in a cross-classification scheme. To determine invariant familiarity, we trained the classifier with familiar faces vs. unfamiliar faces and tested it with familiar names vs. unfamiliar names and vice versa. To determine invariance across modalities, the classifier was trained with familiar faces vs. familiar names and tested with unfamiliar faces vs. unfamiliar names, and vice versa. In total, four different classifications were performed, two each for familiarity and modality. In addition, specific classifications were made to analyze the familiarity of faces and names. Figure 1 illustrates the comparisons.

The MVPA was performed for each subject. The individual maps of the respective accuracy data were averaged across runs and across both directions of cross-classification, and logit transformed, yielding two final maps, one for familiarity and one for modality, respectively. In order to gain valuable insight into the neural representations of these cognitive dimensions, these two sets of maps were submitted to a group-level random effects analysis using a *t*-test (against chance level) and corrected for the effects of multiple comparisons with an FDR threshold of q = 0.05, with the cut-off calculated based on an estimation of empirical null Gaussian distributions [25].

## 3. Results

### 3.1. Cross-Modal Abstract Familiarity Information Found in Areas Traditionally Associated with the Processing of Biographical Information and Emotion

The presence of modality-invariant (abstract familiarity) information was established by using a classifier trained on exemplars from a different modality. This method detected whether accurate predictions (above chance) could be achieved for familiar versus unfamiliar stimuli presented in one modality. Notably, abstract familiarity processing was detected across several clusters situated through the bilateral frontal regions: bilateral precuneus, right insula, right supramarginal gyrus, and left cingulate cortex. The anatomical depiction of the results for this factor is provided by using the Automated Anatomical Labeling (AAL3) atlas in Figure 2 and Table 1.

### 3.2. Found Neural Correlates of Specific Face Familiarity Information

The outcomes of face familiarity classification are presented in Figure 3 and Table 2. This analysis revealed distinct patterns of familiarity processing specific to faces. These patterns were bilaterally observed across multiple high-accuracy clusters (adjacent to the abstract familiarity clusters) in the frontal, orbitofrontal, cingulate, precuneus, as well as in the insula and supplementary motor area. There were also found significant patterns in the left temporal pole, left cuneus, left putamen, left pallidum, right inferior temporal and fusiform gyri.

### 3.3. Challenges in Name Familiarity Classification

We were unable to identify any significant patterns for names after applying FDR correction with the aforementioned threshold. A possible explanation is that the response to names is considerably less intense than the response to faces; therefore, the results from the MVPA analysis are less accurate and noisier. To test this particular possibility, we analyzed two univariate contrasts: all faces versus all names, and all names versus all faces. As shown in Figure 4, several significant clusters survived the established threshold for the first mentioned contrast. However, the contrast in the opposite direction did not show any significant activation, supporting the notion of a lower activation for names.

## 4. Discussion

The sensitivity and flexibility of MVPA for detecting abstract processing permit it to be a valuable approach to complement traditional univariate activation analysis in fMRI studies. In our study, we used a searchlight approach of the MVPA method to determine the way familiarity is represented in the brain. We had two previous hypotheses: (1) one can observe multivoxel patterns for familiarity in different brain areas, reflecting the common properties of the two types of modalities (faces and names), and (2) familiarity can be accurately decoded for a specific modality in some areas, in which no common patterns exist for both kinds of stimuli. The results revealed that whereas certain areas in the brain participate in the processing of familiarity regardless of the type of stimuli, others are specific to the familiarity of faces.

Regarding modality-invariant familiarity analysis, the resulting areas were a large set of frontal regions, traditionally activated by the recognition of famous people and the retrieval of biographical semantic information [17,26,27]: temporal areas, the left supramarginal gyrus, the left precuneus, the superior and middle frontal gyrus, and the left middle cingulate gyrus. We did not replicate the previously observed tendency to left lateralization for this set of structures [28], maybe due to the important contribution of faces to the effect of the abstract familiarity in our data.

In addition to the modality-invariant familiarity areas, we also found regions that seem to process familiarity specifically for faces. The areas processing face familiarity included bilateral clusters adjacent to the abstract familiarity in the frontal, orbitofrontal, cingulate cortex, precuneus, insula, and supplementary motor area, left temporal pole, left cuneus, left putamen, left pallidum, right inferior temporal and right fusiform gyri. This is consistent with other studies that have reported specific areas specialized for face familiarity [17,27]. Originally, Haxby and colleagues’ model [1,2,3] proposed that the areas of the so-called extended system were not specific for faces, but participated in the processing of socially relevant features associated with them, such as pleasantness and familiarity. Our findings, in line with other MVPA studies, supported the existence of a set of regions that are specific for the processing of those features associated with faces. Those are findings that allow us to suggest that our initial hypotheses are not competitive but indeed complementary.

Interestingly, our results also confirmed the growing evidence supporting the involvement of the fusiform gyrus in the processing of face familiarity. The significance of these face-related regions has been underscored by several fMRI studies conducted with prosopagnosic patients [29,30,31]. These studies have consistently demonstrated the pivotal role of these regions in face processing. The evidence from imaging studies on this particular issue comes from investigations that use MVPA methods to identify patterns of familiarity information in the brain [32]. The traditional activation-based methods failed to reliably account for subtle differences in the representation of information of similar stimuli with slightly different characteristics, such as familiar and unfamiliar faces [17,18].

The anterior temporal cortex is another area included in Haxby’s model as part of the extended system. It has been associated with the storage of biographical information. Several fMRI studies have reported face-identity discrimination in anterior portions of the temporal lobes, specifically in the area defined as ATL-FA [33,34,35]. We did not use an fMRI pulse sequence, specially designed to avoid the susceptibility artifact, which is common in this area, causing signal loss. However, we found patterns of information related to specific face familiarity in the left temporal pole. This result confirms the existence of a specific area in the anterior temporal cortex that selectively processes familiar faces.

We were not able to identify any significant pattern for name familiarity using the FDR-corrected threshold. This could be due to the low saliency of names compared to faces. Faces have more relevance for familiarity recognition than names, because of the univocal relation between the identity of the familiar person and its face, and the fact that different persons are named the same. The lower power of the signal for names compared to faces was demonstrated in the univariate contrasts of all faces versus all names and all names versus all faces. The lower magnitude of response for names makes MVPA analysis less accurate and noisier for this experimental condition.

The differences in neural processing between familiar faces and names can be attributed to the distinct cognitive and perceptual mechanisms involved in recognizing these two types of stimuli. Faces are highly salient visual cues, and the human brain is evolutionarily wired to prioritize and excel in facial recognition [36]. The inherent variability in the intensity and nature of responses to visual and linguistic stimuli, such as names, may contribute to differences observed in neural processing [37]. Additionally, the social and emotional significance attached to faces, as primary markers of identity, might evoke stronger and more nuanced neural responses compared to the relatively abstract nature of names [12]. The intricate interplay of sensory, cognitive, and emotional factors underscores the unique neural signatures associated with processing familiar faces and names.

In our study, MVPA was demonstrated to be a useful method for discriminating subtle differences between stimuli, such as the quality of “being familiar” or not. Our results are in line with the growing evidence that using MVPA sustains the existence of some neural areas that seem to process familiarity for stimuli of great salience such as faces. This distinction is possible because MVPA methods analyze patterns of activity across multiple voxels, rather than just focusing on individual voxels, and allow for a higher specificity than activation-based methods in identifying brain regions involved in a particular cognitive process or behavior, with less susceptibility to noise and artifacts in the fMRI signal. This improves the reliability and reproducibility of the results.

Our findings implicate the existence of an integrated set of shared (abstract) and modality-specific areas (specifically for faces) that appear to work in concert with the recognition of familiar people. Because there is evidence suggesting that the processing of visually familiar, famous, and personally familiar faces does not lead to the same neural representation [27,38,39,40], we suggest to continue investigating the influence of different degrees of familiarity in the face familiarity processing network. It could also be significant to perform the cross-classification analysis using other types of contrast stimuli such as voices or bodies.

Our findings may also have practical implications for understanding neuropsychological disorders of face recognition and memory such as prosopagnosia and person recognition disorder. In prosopagnosia, people are unable to recognize familiar faces, although they can recognize familiar people from their voices and names. This suggests that, in people with prosopagnosia, access to semantic memory and knowledge is preserved but this access does not follow the modality-specific route for faces. It must follow either the shared (abstract) areas for faces and names or the modality-specific areas for names. In contrast, in person recognition disorder, patients are unable to recognize familiar people not only by their faces, but also by their voices and names [41]. This suggests that although the modality-specific areas for faces and names might be preserved, the neuropsychological mechanism underlying the person recognition disorder is likely to involve a more central deficit affecting semantic memory and knowledge. This information may help cognitive therapists to design treatment interventions focused on the specific deficits underlying each of these disorders.

## 5. Conclusions

Our findings support the dominant significance of faces in recognizing familiar individuals when contrasted with names. Specifically, we identify neural areas specialized in face familiarity processing. We also find a subset of brain areas accurately decoding familiarity regardless of the input modality. Collectively, these areas seem to contribute to recognizing familiar individuals.

However, the discussion prompts further exploration into the nuanced dynamics of familiarity processing, particularly the influence of varying degrees of familiarity on the activity of these specialized brain areas. To advance our understanding, future investigations could consider alternative experimental designs that involve degrees of familiarity beyond the tested “loved familiar”, and explore the role of other stimuli like voices and bodies in the neural processes underlying recognition. This approach would provide a more comprehensive examination of the intricacies involved in familiar stimuli processing, offering valuable insights into the multifaceted nature of human recognition mechanisms.

## Figures and Tables

**Figure 1 brainsci-14-00039-f001:**
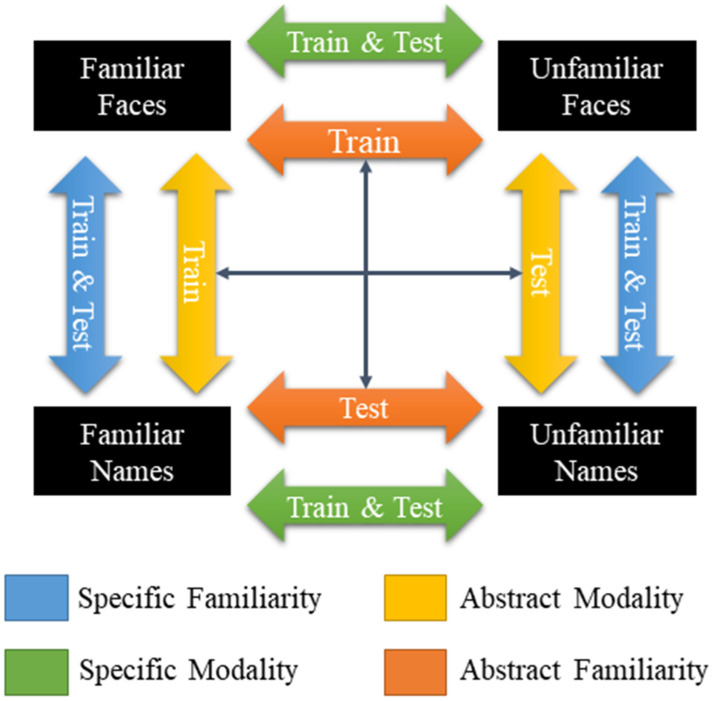
Scheme of the MVPA fast Gaussian Naïve Bayes classifier method. The color legend illustrates the performed comparisons involving the four experimental conditions.

**Figure 2 brainsci-14-00039-f002:**
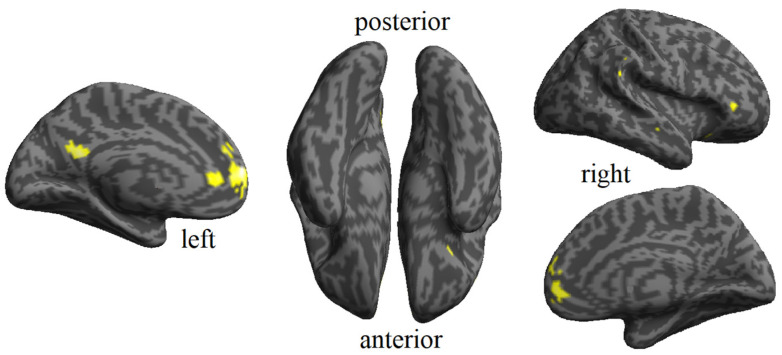
Abstract familiarity map resulting from the cross-validated MVPA for abstract familiarity effect. Clusters are shown using an FDR-corrected q < 0.05.

**Figure 3 brainsci-14-00039-f003:**
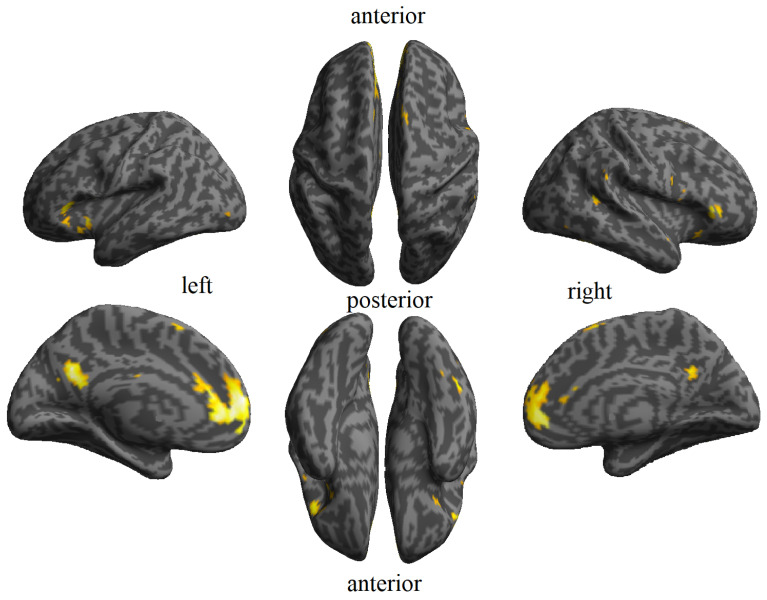
Face familiarity maps resulting from the cross-validated MVPA for abstract familiarity effect. Clusters are shown using an FDR-corrected q < 0.05.

**Figure 4 brainsci-14-00039-f004:**
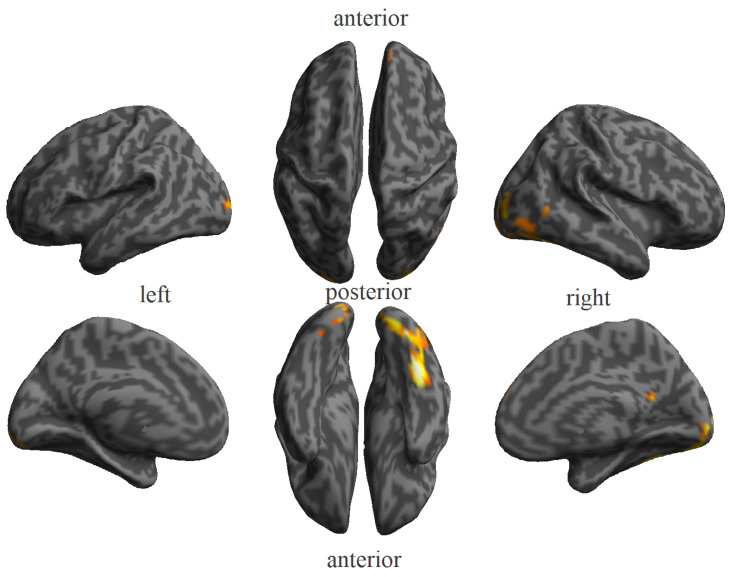
Clusters of activation resulting from the univariate contrast all faces versus all names. Clusters are shown using an FWE corrected *p* < 0.05 (T = 6.007).

**Table 1 brainsci-14-00039-t001:** Clusters resulting from the cross-validated MVPA for abstract familiarity effect. Cluster labelling is shown using AAL3. FDR-corrected q < 0.05.

Area	Coordinates			No. Voxels per ROIs
	x	y	z	
Frontal_Sup_Medial_L	−4	58	12	1304
ACC_pre_L				1304
Precuneus_L	−6	−50	22	167
Cingulate_Post_L				167
Precuneus_R				167
Frontal_Inf_Tri_R	44	38	4	20
Insula_R	32	18	−12	27
Caudate_R	16	22	−10	9
Temporal_Sup_R	44	−18	−10	6
Supramarginal_R	64	−42	26	33

**Table 2 brainsci-14-00039-t002:** Clusters resulting from the MVPA for face familiarity effect. Cluster labelling is shown using AAL3. FDR-corrected q < 0.05.

Area	Coordinates			No. Voxels per ROIs
	x	y	z	
Frontal_Sup_Medial_L	−4	48	8	2365
ACC_pre_L				2365
Precuneus_L	−8	−58	28	456
Cingulate_Post_L				456
Precuneus_R				456
Frontal_Inf_Tri_R	50	32	0	116
Frontal_Inf_Orb_2_R				116
Supp_Motor_Area_R	−4	14	60	112
Supp_Motor_Area_L				112
Temporal_Inf_R	46	52	−12	33
Fusiform_R				33
Insula_L	−42	24	0	475
Frontal_Inf_Tri_L				475
Frontal_Inf_Orb_2_L				475
Frontal_Inf_Oper_L	54	6	22	475
Temporal_Pole_Sup_L				475
Pallidum_L	−18	2	2	60
Putamen_L				60
Insula_R	32	24	−14	61
Frontal_Inf_Orb_2_R				61
OFCpost_R				61

## Data Availability

The data presented in this study are available on request from the corresponding author. The data are not publicly available due to the personal information from experimental subjects included in the images metadata.

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
