# Peer review of "Familiarity Processing through Faces and Names: Insights from Multivoxel Pattern Analysis"

_brainsci, 2023, doi:10.3390/brainsci14010039_

Round 1

Reviewer 1 Report

Comments and Suggestions for Authors

the paper discussing the brain's processing of personal familiarity through searchlight multivoxel pattern analysis (MVPA) of fMRI data contributes interesting insights into the neurobiological basis of familiarity recognition. The identification of specific cortical areas involved in cross-classifying familiar stimuli and the distinction between responses to familiar faces and names are particularly noteworthy. However, there are several areas where the study could be improved or expanded upon for a more comprehensive understanding of the topic.

1. The paper could benefit from a more detailed explanation of the MVPA methodology, particularly the specifics of how the fMRI data were analyzed and interpreted. Information about the sample size and the demographic makeup of the participants is crucial. This would help in understanding the generalizability of the findings.

2. A more in-depth discussion of the statistical methods used to analyze the data and determine significance would strengthen the study. This includes details on the statistical tests employed and the thresholds for significance.

3. While the paper notes differences in the brain's response to familiar faces versus names, a more detailed exploration of why these differences exist would be insightful. Theoretical explanations or references to existing literature could provide a more comprehensive understanding.

4. Placing the findings within the broader context of neurobiological research on memory and recognition would enhance the study’s relevance. This could involve discussing how these results align or contrast with existing theories of memory and familiarity.

5. Addressing the limitations of the study, such as potential biases or confounding factors, is important for a balanced view. Suggesting avenues for future research, such as exploring other types of stimuli or investigating the brain’s response in different populations (e.g., individuals with memory disorders), would be valuable.

6. Practical Implications: Discussing the practical implications of these findings, such as applications in understanding memory disorders or in designing cognitive therapies, would add value to the paper. Including more graphical representations of the fMRI data and the results of the MVPA analysis would aid in visualizing and understanding the complex data.

Comments on the Quality of English Language

Can be improved or taken help from MDPI English editing service. 

Reviewer 2 Report

Comments and Suggestions for Authors

The following comments should be addressed.

1.     The authors claim that the name familiarity could not be identified after using the FDR threshold, I advised the authors to give further visual proof or case learning.

2.     The authors claim to have found regions that specifically deal with face familiarity, and I recommend that the authors design ablation experiments targeting the aforementioned regions to verify the reliability of the conclusions.

3.     The authors’ experimental process only considers familiarity and unfamiliarity, and different degrees of familiarity should also be considered in future studies.

4.     Individuals may not have the same level of familiarity with celebrities, which may affect the experimental results.

Round 2

Reviewer 2 Report

Comments and Suggestions for Authors

All my concerns have been addressed.